# Transcriptome Analysis Identified Coordinated Control of Key Pathways Regulating Cellular Physiology and Metabolism upon *Aspergillus flavus* Infection Resulting in Reduced Aflatoxin Production in Groundnut

**DOI:** 10.3390/jof6040370

**Published:** 2020-12-16

**Authors:** Pooja Soni, Spurthi N. Nayak, Rakesh Kumar, Manish K. Pandey, Namita Singh, Hari K. Sudini, Prasad Bajaj, Jake C. Fountain, Prashant Singam, Yanbin Hong, Xiaoping Chen, Weijian Zhuang, Boshou Liao, Baozhu Guo, Rajeev K. Varshney

**Affiliations:** 1Center of Excellence in Genomics & Systems Biology (CEGSB), International Crops Research Institute for the Semi-Arid Tropics (ICRISAT), Hyderabad 502324, India; p.soni@cgiar.org (P.S.); m.pandey@cgiar.org (M.K.P.); namitasingh9991@gmail.com (N.S.); p.bajaj@cgiar.org (P.B.); 2Department of Genetics, Osmania University, Hyderabad 500007, India; prashantsingam@gmail.com; 3Department of Biotechnology, University of Agricultural Sciences, Dharwad 580005, India; nayaksn@uasd.in; 4Department of Life Sciences, Central University of Karnataka, Gulbarga 585367, India; rakeshgupta.hcu@gmail.com; 5Integrated Crop Improvement, International Crops Research Institute for the Semi-Arid Tropics (ICRISAT), Hyderabad 502324, India; h.sudini@cgiar.org; 6Department of Biochemistry, Molecular Biology, Entomology and Plant Pathology, Mississippi State University, Starkville, MS 39762, USA; jcf416@msstate.edu; 7Guangdong Provincial Key Laboratory of Crop Genetic Improvement, South China Peanut Sub-Center of National Center of Oilseed Crops Improvement, Crops Research Institute, Guangdong Academy of Agricultural Sciences, Guangzhou 510640, China; hongyanbin@gdaas.cn (Y.H.); chenxiaoping@gdaas.cn (X.C.); 8Institute of Oil Crops, Fujian Agriculture and Forestry University (FAFU), Fuzhou 350002, China; weijianz1@163.com; 9Oil Crops Research Institute, Chinese Academy of Agricultural Sciences, Wuhan 430062, China; liaoboshou@caas.cn; 10USDA-ARS, Crop Genetics and Breeding Research Unit, Tifton, GA 31793, USA

**Keywords:** *Aspergillus flavus*, groundnut, aflatoxin production, RNA-Seq, transcriptome analysis, groundnut improvement

## Abstract

Aflatoxin-affected groundnut or peanut presents a major global health issue to both commercial and subsistence farming. Therefore, understanding the genetic and molecular mechanisms associated with resistance to aflatoxin production during host–pathogen interactions is crucial for breeding groundnut cultivars with minimal level of aflatoxin contamination. Here, we performed gene expression profiling to better understand the mechanisms involved in reduction and prevention of aflatoxin contamination resulting from *Aspergillus flavus* infection in groundnut seeds. RNA sequencing (RNA-Seq) of 16 samples from different time points during infection (24 h, 48 h, 72 h and the 7th day after inoculation) in U 4-7-5 (resistant) and JL 24 (susceptible) genotypes yielded 840.5 million raw reads with an average of 52.5 million reads per sample. A total of 1779 unique differentially expressed genes (DEGs) were identified. Furthermore, comprehensive analysis revealed several pathways, such as disease resistance, hormone biosynthetic signaling, flavonoid biosynthesis, reactive oxygen species (ROS) detoxifying, cell wall metabolism and catabolizing and seed germination. We also detected several highly upregulated transcription factors, such as ARF, DBB, MYB, NAC and C2H2 in the resistant genotype in comparison to the susceptible genotype after inoculation. Moreover, RNA-Seq analysis suggested the occurrence of coordinated control of key pathways controlling cellular physiology and metabolism upon *A. flavus* infection, resulting in reduced aflatoxin production.

## 1. Introduction

Aflatoxins are naturally occurring mycotoxins in several important crops such as corn and groundnuts, which serve as the main source of aflatoxin contamination for humans [1]. *Aspergillus flavus* and *A. parasiticus* are two important members of the *A. flavus* group of fungi that often infect groundnut and produce hepatotoxic, carcinogenic, and teratogenic toxins called aflatoxins and have a significant effect on international trade and human health [2,3]. Among the four several known aflatoxins (B_1_, B_2_, G_1_, and G_2_), aflatoxin B_1_ is the most potent aflatoxin, which causes liver cancer by aggravating infection of hepatitis B and C viruses [4].

In groundnut, aflatoxin contamination occurs pre-harvest, during harvesting, and post-harvest mostly during drying, storage and transportation processes [2,5,6]. The aflatoxin accumulation during storage and transport is most devastating, often resulting in the outbreaks of acute aflatoxin poisoning, stunted growth, and malnutrition in children [7]. During post-harvest conditions, pathogen and host interaction will be at the cotyledon tissue stage where aflatoxin is produced [5]. During the post-harvest stages, the *A. flavus* penetrate the groundnut shell and seed coat to colonize the cotyledons to derive sustenance and produce the aflatoxin. The interactions between *A. flavus* fungi and groundnut seed that lead to aflatoxin production during post-harvest, especially in storage, are very important and largely ignored due to lack of reliable screening protocols, and limited understanding of the genetics of resistance.

Interestingly, certain genotypes in groundnut have showed moderate resistance to the fungal infection and aflatoxin production. Three major components of groundnut resistance evaluation include: in-vitro seed colonization (IVSC), resistance to pre-harvest aflatoxin contamination (PAC), and resistance to aflatoxin production (AP) [2,6,8]. Furthermore, there are three types of resistance elaborated in response to aflatoxin contamination: (i) pod infection (shell structure), (ii) seed coat barrier, and (iii) aflatoxin production (cotyledon) [5]. This study elaborates the genetic factors associated with the third level of resistance in cotyledon tissues. In the third level, *A. flavus* is able to infect and colonize the seed coat but will lead to less or limited aflatoxin production. Research has been conducted to identify a genetic basis for the pathogenicity of different isolates of *A. flavus* [9,10]. Transcriptome analyses for the first (IVSC) and second (PAC) resistance mechanisms have revealed the host–pathogen interactions and identified the key candidate transcripts in groundnut [11,12]. However, limited information is available for the third level of resistance i.e., AP resistance [7,13]. In this study, we used two contrasting genotypes for aflatoxin production that differ in their ability to initiate and/or inhibit aflatoxin production at four different sampling times after the inoculation of *A. flavus* in order to gain some understanding of temporal expression of resistance to aflatoxin production.

To gain a better understanding of the AP resistance mechanism, we performed a global transcriptome profiling of two contrast groundnut genotypes by using the RNA-Seq method. Here, we reported the global gene expression between post-harvest seeds of resistant (U 4-7-5) and susceptible cultivar (JL 24) at four different time intervals under the fungus infection. This study identified the host-derived candidate transcripts that control aflatoxin production, which could contribute to the current understanding of the AP resistance mechanism and could be utilized in breeding for resistance to aflatoxin production after thorough validation.

## 2. Materials and Methods

### 2.1. Plant and Fungal Materials

Seeds of the resistant genotype (U 4-7-5) and susceptible genotype (JL 24) were used for conducting the transcriptome study for AP resistance. Characterization of the toxigenic *A. flavus* strain AF 11-4 was done at the Groundnut Pathology Unit of ICRISAT for conducting an RNASeq experiment for AP resistance. The AF 11-4 strain was maintained and sub-cultured on Potato Dextrose Agar (PDA) plates. After 7 days of incubation at 25 °C, conidial suspension was prepared and adjusted to a concentration of (1 × 10^6^ spores/mL).

### 2.2. Aflatoxin Quantification

The total accumulation of aflatoxins was estimated for both control and infected seeds following the protocol suggested by Waliyar et al. [14] using indirect competitive enzyme-linked immunosorbent assay (ELISA). In this, we used polyclonal antibodies that were produced against aflatoxin B_1_ (AFB_1_) for quantitative estimation of total aflatoxins. Aflatoxin B1-Bovine Serum Albumin (AFB_1_-BSA), obtained from Sigma (Catalogue No. 6655, Suffolk, NY, USA), was used for the production of polyclonal antibodies to AFB_1_. Polyclonal antibodies used in the indirect competitive ELISA were produced in New Zealand White inbred rabbit following the protocol developed in our institute previously [15]. The seed coats of the control and infected genotypes were stained with Coomassie Brilliant Blue and were observed through a Florescence microscope (Zeiss Axio Scope.A1, Carl Zeiss, Oberkochen, Germany) to visualize fungal colonization and sclerotia formation.

### 2.3. Screening for Aflatoxin Production

Approximately 200 seeds of both the genotypes were surface sterilized in 0.1% HgCl_2_ for 3–4 minutes and then further washed three times with sterile distilled water for 4 minutes each time. After that two sets of samples (infected and control) were made for both the genotypes. Further, transferring of one set of sterilized seeds i.e., ~100 seeds was done on petri dishes with sterilized filter paper and were used as the control. For another ~100 seeds, the seed coats of resistant (U 4-7-5) and susceptible (JL 24) genotypes were scarified with a sterile needle and inoculated artificially with a mixture of *A. flavus* toxigenic strain AF 11-4 under in vitro conditions. The inoculated seeds were incubated for 7 days by providing congenial conditions for the growth of fungus and AP. Furthermore, a moisture condition with 100% relative humidity at 28 °C in dark was also provided to both the sets. Both the control as well as the infected samples of U 4-7-5 and JL 24 genotypes were taken at day 1, day 2, day 3 and day 7 after inoculation, respectively. In general, aflatoxin production and fungal mycelia growth can happen completely by 7 days after inoculation provided suitable substrate, and growth conditions for the *A. flavus* fungi. The fungal colonization starts within the first 24 h and toxin production initiates after 48–72 h of inoculation and completes by the 7th day. Neither significant growth of fungal mycelium nor aflatoxin production can be seen after 7 days of incubation [16]. Freshly harvested seeds with similar weight (10-g) were used in the case of each cultivar in two replications. About 10–12 seeds at each time interval were frozen in liquid N_2_ until further use and aflatoxin estimation was done for the remaining seeds. In total, there were 16 samples with these combinations (2 genotypes × 4 stages × 2 treatments) which were used for aflatoxin estimation and RNA isolation.

### 2.4. RNA Isolation and Sequencing

Total RNA was extracted from all the 16 samples using “NucleoSpin^®^ RNA Plant” kit (Macherey-Nagel, Duren, Germany) which were collected 7 days after incubation. RNA quality was assessed using Nanodrop 1000 spectrophotometer (Thermo Fisher Scientific Inc., Waltham, MA, USA) and Agilent RNA 6000 Nano chip on Agilent 2100 Bioanalyzer (Agilent Technologies, Palo Alto, CA, USA). The cDNA library construction was performed in an Illumina TruSeq RNA Sample Preparation v2 LS Kit (Illumina Inc., San Diego, CA, USA) according to manufacturer’s instructions. The quantification and size distribution of the enriched libraries were also checked using an Agilent 2100 Bioanalyzer system (Agilent Biotechnologies, Palo Alto, CA, USA) with a High Sensitivity DNA kit. All libraries were sequenced on a NextSeq 500 platform to generate 75 bases paired-end reads at Genotypic Technology Pvt. Ltd. Briefly, 5 μg of the total RNA pooled in equal quantity from two biological replicates were used for the construction of cDNA library using an mRNA-Seq sample prep kit (Illumina Inc., San Diego, CA, USA) following the manufacturer’s guidelines. Subsequently, the RNA samples with a 260/280 ratio of 1.8 to 2.1, 260/230 ratio of 2.0 to 2.3 and RNA integrity number (RIN) value of >8.0, were pooled for paired-end sequencing on the NextSeq 500 platform. The raw reads were subjected to quality filtering using NGS-QCbox [17] and Trimmomatic v0.33 [18] to remove low-quality sequencing reads with ambiguous nucleotides and any adapter contamination. The sequencing data have been submitted to National Center for Biotechnology Information (NCBI) under BioProject ID PRJNA679430.

### 2.5. Read Alignment and Gene Expression Estimation

Post trimming, the filtered reads were assembled into transcripts as per a method adapted from Chen et al. [19] and Clevenger et al. [11]. In brief, the filtered reads were aligned to pseudomolecules of the two diploid progenitor species of cultivated groundnut i.e., *A. duranensis* (A genome) and *A. ipaensis* (B genome) using tophat2 v2.1.1 [20] and bowtie2 v2.2.4 [21]. The alignment files were then separated for reads aligned on A and B genomes and then assembled separately based on a genome-guided approach using Trinity v2.2.0 [22]. Furthermore, the unaligned reads were also assembled using the *de novo* approach. The assembled transcripts at each time point were then filtered for redundancy using the Evidential Gene pipeline [23]. The expression levels of transcripts were estimated in terms of Fragments Per Kilobase of exon per Million Reads Mapped (FPKM) using the Cuffdiff program within Cufflinks v2.2.1 [24]. A transcript was considered to be expressed when FPKM ≥ 1 in at least one sample. For a transcript to be differentially expressed, |log_2_(Fold change)| ≥ 2 and a *p*-value ≤ 0.05 was considered.

### 2.6. Transcripts Annotation, GO Term and Pathway Identification

Transcripts were subjected to BLASTX similarity search with a cut-off of E-value ≤ 10^−5^ against NCBI nr database taxon Viridiplantae, to identify their functions. Gene ontology (GO) annotation and pathway analysis was performed using Blast2GO v5 [25]. Transcripts were further subjected to BLASTX against the PlantTFDB [26] with a cut-off of E-value ≤ 10^−10^ to identify candidate transcription factors. Constitutively expressed transcripts were identified using coefficient of variation (CV), where CV was calculated as the ratio between standard deviation (σ) and mean (µ) for log_2_(fpkm + 1) values for each transcript across the samples (Appendix A). Stably expressed transcripts were identified with their coefficients of variation ≤ 5%.

## 3. Results

### 3.1. A. flavus Infection, Sequencing of RNA Samples and Development of Transcriptome Assembly

The aflatoxin content in both the susceptible (JL 24) and resistant (U 4-7-5) genotypes was estimated using indirect competitive ELISA using the seed coat and cotyledon part of the control and infected seeds. On the first day of infection (ID1), very low mycelial growth was seen in JL 24 and no mycelial growth was observed in U 4-7-5. On second day of infection (ID2), the infected JL 24 began to show profuse mycelial growth and sporulation as compared to U 4-7-5. At ID3 and ID7, very low mycelial growth was observed in U 4-7-5. Uniform germination was seen in both the genotypes in control conditions in all days but the infected JL 24 failed to germinate due to fungal invasion and colonization after infection. Furthermore, at ID7, the highest toxin production, 43,989.6 (µg/kg) was observed in JL 24 (Figure 1). To perform RNA-Seq, a total of 16 samples were collected at four post-infection time points (24 h, 48 h, 72 h and 7th day after inoculation) from U 4-7-5 (resistant) and JL 24 (susceptible) genotypes and their controls. RNA-Seq generated 840,453,856 reads with an average of 52,528,366 reads per sample (Table 1). Subsequently, a transcriptome assembly comprised of 74,026 transcripts with total size of ~81.59 Mbp and N50 of 1626 bp was developed (Table 2). The average size of the assembled transcript was 1102 bp with a maximum transcript size of 12,681 bp. After a stringent quality check, an average of 83.67% reads per sample were aligned to the assembled transcripts.

### 3.2. Differential Gene Expression in Control and Infected Samples

The expression level, FPKM (fragments per kilobase million reads mapped), of each gene was estimated based on RNA-Seq data for all the 16 samples collected from control and infected samples of JL 24 and U 4-7-5 with the transcriptome assembly as mentioned above. To exclude transcripts with low confidence expression value, only transcripts with an FPKM value ≥1 in at least one of the samples were assigned as expressed [27]. Based on the above criteria, 67,812 (91.60%) transcripts out of 74,026 were identified as transcriptionally active (Appendix A). Among the 16 samples, ID 2-U 4-7-5 (second day of infection) followed by ID7-JL 24 (7th day of infection) had the highest number of expressed genes. On the other hand, the largest number of highly expressed transcripts (FPKM >20) were observed in the CD1-U 4-7-5 (first day of control) (10,777 transcripts), respectively. Subsequently, the smallest number of highly expressed genes (FPKM >20) were observed in the ID-7 JL 24 (9029 transcripts) (Figure 2). In the moderate category 2 ≤ FPKM < 20, the largest number of highly expressed transcripts 2 ≤ FPKM < 20 were observed in the CD7-JL 24 sample (36174 transcripts). In the low category 0 ≤ FPKM < 2, the largest number of highly expressed transcripts 2 ≤ FPKM < 20 were observed in the ID3-JL 24 sample (19,753 transcripts).

A total of 1779 unique differentially expressed genes/transcripts (DEGs) were identified by comparing the 16 samples (Appendix A). Upon considering DEGs at four different time points in two contrasting cultivars, a higher number of DEGs (2740) showed expression in U 4-7-5 as compared to JL 24 (2554). We found 1866 unique DEGs in U 4-7-5 and 1634 unique DEGs in JL 24, respectively. Specifically, a high number of DEGs were observed at 1 ID (1178) and 3 ID (982) in U 4-7-5, while in JL 24 a high number of DEGs were observed at 1 ID (910) and 7 ID (765). However, the resistant genotype, U 4-7-5, which showed highly differentially expressed genes was at 1 ID and 3 ID, but the susceptible genotype did not show a similar level of expression pattern.

### 3.3. Functional Annotation and Pathway Assignment

Gene ontology (GO) annotation was performed to assign a function to differentially expressed transcripts and their products. The functions of DEGs were characterized based on three ontologies namely molecular function, cellular component and biological processes (Figure 3). Among biological processes, the DEGs of cellular processes, metabolic processes and single organism processes were the most pre-dominant ones. In the cellular component category, 26,092 unigenes were involved in the cell part, cell, membrane part etc. (Figure 3). In the molecular function category, the largest number of transcripts were assigned to catalytic activity (7864), and binding (7754). Several transcripts were also involved in categories like signal transduction mechanisms, structural molecular activity, molecular transducer activity, electron carrier activity, and antioxidant activity. The majority of transcripts were mapped to 143 pathways in the KEGG (Kyoto Encyclopedia of Genes and Genomes) database related to the biosynthesis of antibiotics (124), purine metabolism (35) and amino sugar and nucleotide sugar metabolism (32) (Appendix A).

### 3.4. Transcription Factors Associated with AP Resistance Mechanism

Transcription factors (TFs) are the molecular switches that play a critical role in plant development and defense. In the present study, a total of 55 families of TFs were identified and the largest number of transcription factors were from TF families: bHLH (994), MYB-like (632), NAC (609), ERF (463), bZIP (446) (Appendix A). These TFs could be assigned to play an important role in many cell physiological and biochemical processes and also could be involved in the cross-talk between plant–pathogen interactions, which showed the regulation of plant resistance for the pathogen. Among the 36 differentially expressed TF encoding transcripts, the highest number of transcripts were from the WRKY and MYB TF families (Appendix A). Moreover, the analysis revealed the significant upregulation of two transcripts (AP_23518, AU_1429) of the WRKY TF family in the JL 24 upon infection as compared to U 4-7-5. In contrast, one transcript (UN 4898) of the WRKY TF family was highly upregulated in U 4-7-5 compared to JL 24 upon infection. Most importantly, another transcript (AP_17045) of the bHLH encoding gene family was highly upregulated in JL 24 upon infection across the stages compared to U 4-7-5. Another transcript (UN_4370) of the bHLH encoding gene family was highly upregulated after 24 h in case of U 4-7-5 as compared to JL 24. In U 4-7-5 genotype, upon infection, a few TFs families such as ARF, DBB, MYB, NAC and C2H2 were found to be highly upregulated compared to JL 24 upon infection.

### 3.5. DEGs Associated with AP Resistance Mechanism

During aflatoxin contamination, the first 72 h are crucial as they involve seed colonization by *A. flavus* fungus and toxin production by hijacking host machinery and suppressing the host defense mechanism. *A. flavus* growth significantly increases after infection on seed and sporulation becomes visible within 48 h and afterwards aflatoxin production (AP) by fungus drastically increases and continued up to the 7th day (Figure 1). Therefore, the groundnut resistance to AP mainly depends on the transcripts which are differentially expressed upon initial stages of infection and restrict both fungus growth and toxin production. The AP mechanism activates several resistant transcripts/pathways/transcription factors that induce resistance against pathogen attack (Figure 4).

#### 3.5.1. Expression of Disease Resistance-Related Transcripts

Analysis of resistant and susceptible genotypes in response to AP by *A. flavus* depicted a complex expression pattern among the key defense-related transcripts. We observed that two transcripts (AP_235 and AU_663), encoding stress-responsive annexin D1 protein were drastically increased up to 48 h in the U 4-7-5 seeds after infection. Most importantly two transcripts (AP_4682 and AU_19005) encoding disease resistance protein (leaf rust 10 disease-resistance locus receptor-like protein kinase-like) were significantly upregulated for 48 h in both control and the infected seeds of U 4-7-5. Intriguingly expression of AP_4682 was only detected in the seed sample of U 4-7-5 and these transcripts were either absent or not significantly upregulated in JL 24 (Figure 5A) Likewise, during initial 72 h we observed significant upregulation of two transcripts (UN_1201, and AP_10642) encoding disease resistance *RGA1* in the U 4-7-5 compared to JL 24 upon infection (Figure 5A). Intriguingly expression of UN_9417 encoding disease resistance *RGA1* was detected in the seed sample of U 4-7-5 up to 48 h. Additionally, upregulation of transcripts (AU_21237, UN_14094, AP_21088, UN_14095 and AU_16427) encoding (R, S)-reticuline 7-O-methyltransferase, an alkaloid biosynthetic pathway protein, was observed in both JL 24 and U 4-7-5; however, the fold change was dramatically higher in the U 4-7-5 infected seed during the initial 48 h after *A. flavus* infection (Figure 5B). Furthermore, *A. flavus* infection drastically increased the expression of four transcripts (AP_13750, AU_7107, AP_13751 and AU_7106) within 48 h, encoding for carotenoid cleavage dioxygenase 8 homolog chloroplastic protein (*CCD8B* gene) in U 4-7-5 compared to JL 24 (Figure 5A). The *A. flavus* infection severely induced expression of transcripts (UN_5421, UN_7651, AU_13814, UN_5214, UN_5216 and UN_7652) encoding linoleate 9S-lipoxygenase 1 protein (*LOX*); however, their abundance was substantially higher in the U 4-7-5 contaminated seeds (24, 48 and 72 h samples) compared to the JL 24. More interestingly, we observed down regulation of three transcripts (AU_22837, UN_5904 and UN_8861) encoding linoleate 9S-lipoxygenase 1 in the seeds of JL 24 upon infection, whereas these were drastically enhanced in the U 4-7-5 contaminated seeds (24 h, 48 h and 7 days samples) upon infection (Figure 5A). In the infected seed of resistant genotype U 4-7-5, the induced fungal cell wall dissolution transcripts were much more significant compared to the that in the susceptible genotype, such as endochitinase 3, endochitinase-like, endoglucanase 17, beta-galactosidase 3/12/46 etc. (Figure 5B).

Our analysis identified the expression of three transcripts (UN_7595, AU_10684 and UN_4853) encoding inositol-3-phosphate synthase protein (ISYNA1) and one transcript (AP_7685) encoding for myo-inositol-1-phosphate synthase protein (MIPS2) that were drastically induced in the first 48 h upon infection in U 4-7-5 compared to JL 24. The slight increase in the abundance of these three transcripts in JL 24 after infection at 24 h stage was drastically reduced in the later three stages (48 h, 72 h and 7 days) (Figure 5A). Mannose glucose-binding lectin (MBL) is an important pattern recognition molecule for the plant innate immune system and they play an important role in the first hours/days of any primary immune response to the pathogen, we identified nine such MBLs showing significant upregulation in the first 48 h of the infection in the resistant U 4-7-5 (Figure 5B). One of the important findings during data analysis was the identification of response to low sulphur 2, known to positively regulate plant defense [7]. Our data suggested that resistant genotype U 4-7-5 seeds were able to maintain a high level of transcript for the protein response to low sulphur 2 (UN_7434); in fact, upon infection, their abundance was slightly increased at 24 h and 72 h after infection. In contrast, their abundance in the JL 24 infected seeds was substantially lower (four-fold) as compared to the corresponding stages (24 and 72 h) in infected seeds of U 4-7-5 (Figure 5B). Likewise, a protein senescence-induced receptor-like serine threonine-kinase encoded by SIRK gene (UN_2317) known to play an important role in the innate immunity appeared to be expressed consistently in all the stages of both control and infected seeds of U 4-7-5. On the contrary, this was induced late in the 48 h infected seeds of JL 24, then disappeared at the 72 h. Furthermore, its expression was not detected in the 24-h control sample, however, was present at a very low level in the infected seed (24 h) (Figure 5B). Furthermore, well-known abiotic stress (hypoxia) regulator cysteine oxidase was severely downregulated in the U 4-7-5 seeds compared to JL 24 in which their expression was significantly enhanced after *A. flavus* infection (Figure 6B).

We observed early induction of several transcripts involved in the synthesis of oxylipin and other antifungal compounds in the infected seeds of U 4-7-5 such as peroxygenase and epoxide hydrolases. One of the proteins, cathepsin B-like protease 2 (UN_5410), plays a central role in programmed cell death (PCD) associated hypersensitive response (HR) involved in disease resistance [28] and abiotic response in plants. Interestingly, the U 4-7-5 infected seeds not only showed upregulation of cathepsin B-like protease, in fact, its abundance was also substantially higher in the control sample of U 4-7-5 as compared to JL 24 control seeds in two stages (24 h and 7 days). Furthermore, upon infection, two important proteins, Raf kinase inhibitor family (AP_24420) and lipid transfer EARLI 1-like (AP_4299) showed high expression in seeds of the resistant genotype U 4-7-5 (Figure 5A), which are key regulators of the plant defense system and provides innate immunity [29,30]. Among them, AP_24420 showed high expression in seeds of the resistant genotype U 4-7-5 (24 h, 7days) samples and AP_4299 showed high expression in seeds of the resistant genotype U 4-7-5 (24 h, 48 h, 7days) samples. To our surprise, a few DEGs encoding bidirectional sugar transporters (SWEET14, SWEET6b-like and NEC1-like) and DMR6-LIKE OXYGENASE 1 encoded by the DLO1 gene, are best known for suppressing plant immunity and enhancing susceptibility against a pathogen [31,32]. In contrary, we observed significant up-regulation of these genes in the *A. flavus* infected seed of U 4-7-5 (up to 48 h) (Figure 5B).

#### 3.5.2. Differential Expression of Hormone Biosynthetic and Signaling Transcripts

We found several DEGs from hormone signaling and biosynthesis pathways. In resistant genotype U 4-7-5 seeds (24 h, 48 h samples), infection induced the expression of cytokinin biosynthetic transcripts (AU_17993, AP_19178 and AP_5509) through high upregulation and these transcripts encode a key protein cytokinin riboside-5-monophosphate phosphoribohydrolase. Among them, AU_17993 and AP_5509 also showed expression in U 4-7-5 seeds (7 days sample). In contrast, upon infection JL 24 seeds exhibited severe upregulation of six transcripts which encode cytokinin dehydrogenase 2, a key enzyme involved in the cytokinin degradation. The induction of cytokinin dehydrogenase 2 protein was concordant with the constitutive higher expression of salicylate biosynthetic transcripts coding for salicylate carboxymethyltransferase and salicylic acid-binding 2 in the U 4-7-5 control and infected seeds compared to JL 24 (Figure 6A). Upon infection, the expression of transcripts (UN_3486, UN_9693, AP_26238 and AP_26237) encoding for alpha-dioxygenase 1 were either unregulated or remained higher gradually in all stages in the seeds of U 4-7-5 compared to JL 24. Similarly, expression of JA-amino acid synthetase encoded by *JAR1* gene (AP_6282) was drastically increased in the *Aspergillus* contaminated U 4-7-5 seeds compared to the JL 24 in all stages. Expression of jasmonate (JA) biosynthetic proteins allene oxide synthase (AP 6160) was observed in contaminated U 4-7-5 (24 h, 48 h samples) (Figure 6A). As anticipated, expression of six transcripts (AP_21068, AU_583, AU_584) encoding 9-cis-epoxycarotenoid dioxygenase (Abscisic acid biosynthesis key enzyme) were drastically increased upon infection and were concordant with upregulation of stress-induced protein abscisic stress-ripening 2 (AP 26968) and abscisic stress-ripening 3 (AU_26733), whose transcripts were also significantly higher in the seeds of U 4-7-5 after infection (after 24 h), while they were absent in the JL 24. Likewise, expression of transcripts for the late embryogenesis abundant 2 protein encoded by *LEA2* gene was drastically increased in U 4-7-5 seeds upon infection, whereas the expression of *LEA 14* transcripts was increased in both genotypes, but their abundance in U 4-7-5 infected seeds (all stages) were substantially much higher than that of JL 24. Similarly, three transcripts (AP_11635, AU_14183 and UN_2723) for another class of LEA (11kDa *LEA*) protein were highly expressed in U 4-7-5 seeds at the 24-h stage, while these were severely reduced in the JL 24 seeds (Figure 6A).

Furthermore, upon infection, the U 4-7-5 seeds showed drastic downregulation of ABA receptors encoding transcripts for the Abscisic acid receptor protein encoded by the *PYL4* gene (AP_1867, AU_1511, AU_24981 and UN_14990) in the first 24 h, and dramatic upregulation of transcripts (AU_20012, UN_8531 and AP_18058) encoding for transcription factor ABA-INSENSITIVE 5 (*ABI5*) in the first 24 h. ABA is known to have an antagonist interaction with ethylene. We also identified the antagonistic expression pattern of key ethylene biosynthetic protein ACC-oxidase (*AC0-1*) and the *ACC synthase* gene. For instance, we found five transcripts (AP_3601, AU_4670, UN_5227, AU_28502 and AP_29260) of ACC-oxidase and two transcripts (AU_28479 and AU_28480) of the *ACC-synthase* gene were drastically reduced in the infected U 4-7-5 seeds sample (24 h) compared to JL 24. The abundance of these transcripts was gradually increased up to 72 h, while their abundance remained low in JL 24 samples (Figure 6A).

#### 3.5.3. Expression of Key Transcripts Controlling Flavonoids and ROS Detoxifying Transcripts

One of the key mechanisms to restrict pathogen invasion is accumulating antioxidant flavonoids and cell wall lignin by host plants. As anticipated, the expression of several flavonoids’ biosynthetic transcripts encoded proteins were upregulated upon infection in the resistant genotype U 4-7-5 such as chalcone synthase, chalcone isomerase, isoflavone 2-hydroxylase, isoflavone reductase, NAD(P)H-dependent 6-deoxychalcone synthase, isoflavone-7-O-methyltransferase 9-like, UDP-glycosyltransferase 83A1, UDP-glycosyltransferase 87A1-like, etc. Most importantly, the resistant genotype U 4-7-5 exhibited tremendously very high expression for transcripts encoding proteins UDP-glycosyltransferase 83A1 and UDP-glycosyltransferase 87A1-like (in plants they catalyze the last steps of anthocyanin pigment biosynthesis and in mammals they are involved in the detoxification and elimination of toxins and carcinogens) for the first 72 h (Figure 6B). In contrast, seeds of the susceptible genotype JL 24 seeds exhibited no change for the first 24 h, then a severe reduction in transcripts at 48 h and a slight down-regulation at 72 h.

We identified early induction and upregulation of several detoxifying transcripts in the infected seeds of U 4-7-5 compared to susceptible genotype JL 24. These include homologs of peroxidases, glutathione S-transferase, superoxide dismutase (Cu-Zn), cytochrome P450 oxidase, ascorbate peroxidase. Notably, the expression of many of these proteins was significantly downregulated in the first 24 h seeds in JL 24 upon infection. The expression of many of these transcripts was significantly higher in U 4-7-5 control samples after 24 h when compared with JL 24 such as glutathione S-transferase (AU_23379) and ascorbate peroxidase (AU_9550). More importantly, seeds of resistant genotype exhibited consistently higher expression of a Vitamin-E synthesizing gene tocopherol O-chloroplastic isoform X2 (UN_4174) for 72 h upon infection, while JL 24 showed a slight increase at 24 h, but later its expression was severely reduced in the next three stages (72 h to 7th day) (Figure 6B).

#### 3.5.4. Altered Expression of Transcripts Involved in the Cell Wall Metabolism

The plant cell wall is the main barrier that a fungus has to breach for its colonization. We observed that the transcripts for several lignin biosynthetic transcripts were induced by *A. flavus* infection in U 4-7-5 such as cinnamoyl-reductase 1, caffeic acid 3-O-methyltransferase, caffeoylshikimate esterase, caffeoyl-O-methyltransferase 5, spermidine hydroxycinnamoyl transferase, 4-coumarate—ligase 2 and its isoforms 4-coumarate: coenzyme A ligase and 4-coumarate—ligase-like 7 isoform X2. Likewise, transcripts for the different homolog of gene *XTH* encoding protein xyloglucan endotransglucosylase hydrolase (a key enzyme of cell wall xylan biosynthesis were upregulated in the U 4-7-5 upon infection in the first 48 h. Additionally, three transcripts (AP_21324, AU_21069 and AU_21070) related to UDP-xylose transporters, which participate in xylan biosynthesis, were found significantly abundant in the U 4-7-5 during the first 48 h after *A. flavus* infection. Notably, we observed upon infection with *A. flavus* that five transcripts (AU_21886, AP_23107, AP_26687, AU_27602 and AP_26688) which encode for repetitive-proline-rich cell wall protein (a cell wall strengthening protein) were drastically increased for the first 48 h in U 4-7-5 seeds (Figure 7A). In contrast, these transcripts were reduced for the first 72 h (except, AU_21886 and AP_23107 in JL 24, which showed slight upregulation at first 24 h). Additionally, we observed transcripts encoding proteins related to cell wall catabolizing transcripts—beta-cell wall isozyme, beta-xylosidasealpha-L-arabinofuranosidase 1, pectinesterase 2 and expansin-A8, were induced in JL 24 upon infection for the first 24 h (Figure 7B).

#### 3.5.5. Influence of *A. flavus* Infection and Toxin Production on Seed Germination

The influence of *A. flavus* infection and AP on seed germination is one of the important observations of the phenotyping assay. The seeds of JL 24 were merely able to germinate at the 7th day after infection with *A. flavus*. We identified important transcripts associated with seed germination and embryo growth that were differentially expressed upon infection in the seeds of JL 24 and U 4-7-5. One of them, *DELAY OF GERMINATION 1-like* (*DOG 1-like; AU_24829*), a negative regulator of germination, was severely reduced (up to 72 h) in the infected seeds of the U 4-7-5 as compared to its control, whereas, in JL 24, for the first 24 h infection caused upregulation of the *DOG 1-like* transcript; later it was reduced. Further, the expression of transcripts associated with zinc finger CONSTANS-LIKE 4 proteins was upregulated upon infection in the U 4-7-5 for first 24 h and were maintained at 48 h, while JL 24 seeds did not show any change and their expression was drastically low when compared with the infected seeds of the U 4-7-5 (24 h) (Figure 7C). In conclusion, we represented highly differentiated 16 transcripts which were upregulated and downregulated at each day (Table 3).

## 4. Discussion

Several studies have been undertaken to identify and dissect the expression of saprophytic fungus *A. flavus* to understand the genetic basis for the pathogenicity [2,6]. However, clarity could not be achieved on host-derived resistance mechanisms dissecting pre-harvest resistance mechanism in the developing seeds in pre-harvest condition or during in-situ seed colonization in groundnut [11,12,33]. Furthermore, the lack of stable resistance sources in the cultivated gene pool and paucity of genomic resources have limited such efforts to understand host-defense mechanism against post-harvest AP through transcriptomics approaches in groundnut [7,13]. In recent years, large scale genomic resources including a large scale of molecular markers [34], genetic maps [35], genome sequence assemblies [36,37], gene expression atlases [38,39] and identification of resistant genotypes for different components of the aflatoxin mechanism [8] has accelerated efforts to understand host-defense mechanisms in groundnut.

This study used U 4-7-5, the resistant parent which is an excellent genetic material for breeding for resistance to post-harvest aflatoxin production resistance. This genotype offers potential for characterization of candidate transcripts and molecular pathways associated with the host-derived resistance against AP. Therefore, we carried an in-depth transcriptome analysis of the contaminated seeds of resistant (U 4-7-5) and susceptible (JL 24) groundnut genotypes at different courses of time post *A. flavus* infection. The RNA-Seq study enables comprehensive and rapid acquisition of global transcript expression in an organ/tissue of a species. As a result, we identified a large number of transcripts (74,026) that were expressed in the *A. flavus* infected seeds; among them, 65,502 were annotated successfully against NCBI nr database taxon Viridiplantae.

In recent years, high throughput sequencing technologies, transcriptome and proteomics studies have been explored to understand the mechanisms associated with aflatoxin contamination in groundnut [2,6]. The host–pathogen interaction is important for the establishment of fungal infection/or to achieve resistance against fungus by a host. The capturing of transcriptomic changes which occurred during the initial phase of infection is key to understand the resistance mechanism as the process of seed colonization starts by germination of spores and mycelial growth occurs in susceptible genotype after inoculation/infection within 24 h, followed by the formation of spores and sclerotia (which are the center for AP) within 48 h. The sclerotia formation drastically increased in 72 h (day 3) and continued for the next few days. Most importantly, we observed upregulation of several disease resistance transcripts encoding proteins for linoleate 9S-lipoxygenases, carotenoid cleavage dioxygenase homolog, disease resistance RGA1, annexin D1 protein, leaf rust 10 disease-resistance locus receptor-like protein kinase-like, RESPONSE TO LOW SULFUR 2, senescence-induced receptor-like serine-threonine-kinase, etc., in resistant genotype U 4-7-5 during the first 24 h or 48 h upon infection. The linoleate 9S-lipoxygenase, peroxygenase and carotenoid cleavage dioxygenase are involved in the stress signaling. A study by Grebner et al. [40] also showed that enzymes linoleate 9S-lipoxygenases and peroxygenase play a key role in inducing stress signal transduction and JA to confer resistance against biotic stress [32]. Carotenoid oxidation products β-cyclocitral also act as stress signaling molecules and activate plant defense mechanism by modulating reactive oxygen species (ROS) machinery cellular defense against stress [41]. During infection, RESPONSE TO LOW SULFUR 2, annexin D1 protein and disease resistance RGA1 are the key controllers for defense responses in plant (https://www.uniprot.org/uniprot/Q9FIR9) [42,43] (Table 4). Notably, the role of annexin protein in abiotic stress management is well characterized [44]; however, its role in biotic stress is yet to be examined. Likewise, upregulations of transcripts encoding the Raf kinase inhibitor family, lipid transfer EARLI 1-like, senescence-induced receptor-like serine threonine-kinases, cathepsin B-like protease 2 and mannose glucose-binding lectin were involved in the innate immunity and PCD [28,29,30,45,46]. This indicates that these transcripts might also play a central role in the conferring resistance to U 4-7-5 against aflatoxin contamination. Furthermore, in the model plant *Arabidopsis,* involvement of myo-inositol metabolism with PCD and basal immunity is well demonstrated. Reduced levels of inositol can induce severe cell death and disease symptoms by suppressing/knockdown of inositol-3-phosphate synthase [47] and myo-inositol-1-phosphate synthase [46]. Wild type phenotype can be restored by feeding these inositol deficient mutants or transgenics with inositol/galactinol [47,48,49]. We also observed upregulation of these key transcripts in the resistant genotype upon infection, being involved in the inositol biosynthesis and metabolism pathways. Furthermore, the impact of the *Aspergillus* infection, growth and toxin accumulation in seeds could severely affect the seed germination procedure by affecting the expression of transcripts such as *DOG 1-like* and *CONSTANS-LIKE 4.*

This study represents early induction of several ROS detoxifying/antioxidant transcripts coding proteins such as peroxidases, glutathione S-transferase, superoxide dismutase (Cu-Zn), cytochrome P450 oxidase and ascorbate peroxidase, in resistant genotype U 4-7-5 upon *A. flavus* infection. Previous studies indicate upregulation of these detoxifying transcripts as important to mitigating aflatoxin contamination and are the key features associated with groundnut resistant genotypes [7,12,13,52]. Generation of ROS, ROS detoxifying enzymes, hypersensitive reaction (HR), and PCD are associated with the plant defense mechanism [53].

Overexpression of antioxidant transcripts enhances resistance/tolerance against both biotic and abiotic stress responses in crop plants [54]. Notably, the expression of multiple transcripts involved in the flavonoids/anthocyanin/polyphenols biosynthesis were upregulated in the U 4-7-5 seeds during *A. flavus* infection. Previously, it has been shown that the presence of these compounds could help in mitigating aflatoxin contamination in groundnut [55]. Likewise, genetic manipulation of crops to improve the flavonoids/anthocyanin/polyphenols content has been broadly used to enhance the antioxidant content and reduce susceptibility to diseases (biotic stress) including fungal diseases [56,57,58,59]. Furthermore, application of flavonoids/polyphenol can significantly reduce the growth of *A. flavus* and toxin production [55,60,61]. Some recent evidence supports the subtle differences in the expression of flavonoids/anthocyanin biosynthetic transcripts between aflatoxin contamination resistant and susceptible genotypes of groundnut [7,12,13] and maize [62].

The *A. flavus* infection in U 4-7-5 induced expression of transcripts encoding proteins such as endochitinase 3, endochitinase-like, endoglucanase 17 and beta-galactosidase 3/12/46 which are responsible for degradation of the fungal cell wall, corroborated by suppression of transcripts involved in the plant cell wall degrading or metabolizing such as beta-cell wall isozyme, beta-xylosidase Alpha-L-arabinofuranosidase 1, pectinesterase 2 and expansin-A8. Furthermore, the resistant genotype induces expression of cell wall biosynthesizing transcripts encoding proteins which are involved in the lignin and xylose enrichment and cell wall strengthening such as cinnamoyl-reductase, caffeic acid 3-O-methyltransferase, caffeoylshikimate esterase, caffeoyl-O-methyltransferase, spermidine hydroxycinnamoyl transferase, 4-coumarate-ligase, xyloglucan endotransglucosylase hydrolase, UDP-xylose transporters and repetitive-proline-rich cell wall protein, this was concomitant with the enhanced expression of MYB6 transcription factor. Some earlier studies also indicate that fungus invasion and mycelial growth into the host tissue involves the dissolution of the host cell wall [63]. In several resistant crops, lignin accumulation during fungal infection and gradual reduction in the expression of cell degrading transcripts is the key method which plants use as defense mechanism [64], including groundnut exhibiting enrichment of lignin biosynthetic transcripts and upregulation of *MYB* gene coherently [7,12,13]. Notably, U 4-7-5 seeds exhibited upregulation of multiple transcripts and their homologs involved in the fungus cell wall degradation i.e., endochitinase 3, endochitinase-like, endoglucanase 17, beta-galactosidase 3/12/46. Furthermore, resistance to fungal disease in crop plants is enhanced by the expression of fungus cell wall catabolizing gene *chitinase* in groundnut [65], tomato [66] and other important crops.

In this study, we also identified transcript alteration associated with cytokinins and ABA biosynthesis and response. The resistant genotype U 4-7-5 showed differential expression of transcripts encoding proteins, cytokinin riboside-5-monophosphate phosphoribohydroxylase and cytokinin dehydrogenase 2, involved in synthesis or degradation of cytokinins. Cytokinins have been previously described to be involved during host–pathogen interaction as well as resistance [67]. Further, cytokinins signaling mediated pathogen resistance could be salicylate (SA)-dependent mechanism, as overexpression of *uni-1D* mutation induces accumulation of both SA and cytokinin [68]. The resistant genotype U 4-7-5 seeds show a constitutive higher abundance of abundance for transcripts encoding salicylate carboxymethyltransferase and salicylic acid-binding 2. The salicylate carboxymethyltransferase involved in the synthesis of methylsalicylate (MeSA) and salicylic acid-binding 2 is required to convert MeSA to SA during defense response to stresses [69,70]. Our study also reflects that RNA-Seq data of the resistant genotype U 4-7-5 showed drastic upregulation of allene oxide synthase and JA-amino acid synthase upon infection. Furthermore, another such study [71] also concluded that endogenous cytokinins can induce the expression of JA as a defense response.

In the resistant genotype U 4-7-5, we also observed initial mycelial growth but the final growth of fungus on the colonized seeds and toxin production was significantly reduced. Furthermore, resistant genotype U 4-7-5 seeds showed early induction of a 9-cis-epoxycarotenoid dioxygenase—a key protein involved in the ABA biosynthesis, and stress or ABA-induced transcripts abscisic stress-ripening 2/3 and *LEA*. The role of ABA during plant–pathogen interaction is complex and demonstrated to have a negative impact on the plant resistance against pathogens in several crop plants such as barley [72], tobacco [73]. Recently, Xu et al. [74] reported ABA can induce initial growth of mycelia in *A. nidulans*; however, it reduces the final biomass by slowing-down cell cycle progression through altering the expression of transcripts involved in the cell cycle and altering secondary metabolite production. The *LEA* gene promoter contains ABA response (ABRE) elements, and transcripts containing these elements in the promoter region were induced by ABA [75]. Additionally, these changes were associated with ABA receptor PYL4 and transcription factor ABA-INSENSITIVE 5 in resistant genotype U 4-7-5. PYL4 is known to be involved in ABA signaling and controls ABA sensitivity via interaction with PHOSPHATASE 2CA and also regulates the JA signaling pathway [76], a key element involved in host–pathogen interaction and host defense.

*ABI5* upregulation has been found to increase the sensitivity to ABA and sugars [77], and the knockout mutants remain ABA insensitive and show altered expression of ABA biosynthetic transcripts [51]. The previous study has demonstrated ABI5 as a key controller of ABA inducible *LEA* transcripts in Arabidopsis [51]. In maize and tobacco, LEA proteins were identified as a regulator of abiotic stress tolerance which reduces the accumulation of reactive oxygen species [78]. In groundnut, a homolog of abscisic acid insensitive, *ABR1*, has been identified as a repressor of the ABA signaling pathway which confers resistance against pre-harvest aflatoxin contamination [11]. Furthermore, an enhanced level of ABA can induce ethylene biosynthesis and production or vice-versa [79]. We also observed a similar trend in the resistant genotype U 4-7-5-infected seeds i.e., a correlation between the upregulation of ABA biosynthetic transcripts and increased expression of ethylene biosynthetic transcripts. These observations suggested an AP resistance mechanism in groundnut involving a complex interaction between phytohormones ABA, cytokinins, JA, SA, and ethylene. Additionally, our finding provokes the question about possible role of ABA during *A. flavus* contamination through signaling the activation of the resistance mechanisms against *Aspergillus* growth and AP. This might be due to the reduction in biomass of fungus along with reduced secondary metabolite production by activity of ABA as indicated in earlier reports of Xu et al. [74].

## Figures and Tables

**Figure 1 jof-06-00370-f001:**
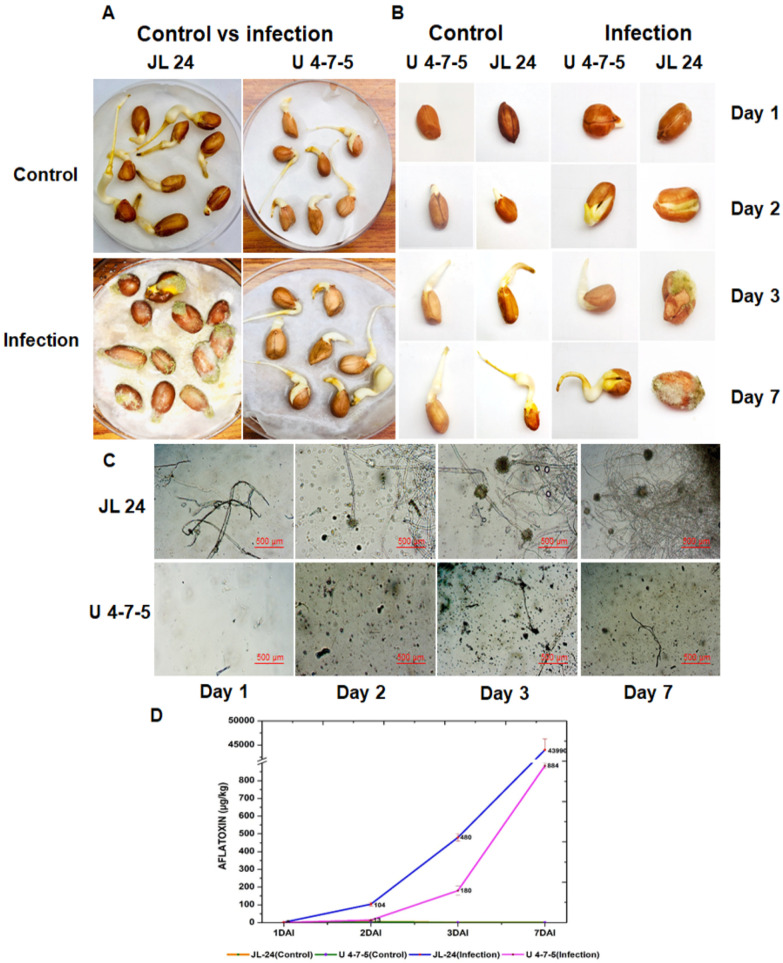
Phenotypic observations for seeds of U 4-7-5 and JL 24 during aflatoxin production (AP) by *A. flavus* at different time points along with microscopic observation and aflatoxin estimation. This figure shows the diagrammatic representation of phenotypic observations for seeds of JL 24 and U 4-7-5 at different time points (**A**,**B**), microscopic observation during AP at different time points (**C**) and graphical representation of AP estimation at different time points under control and infection conditions clearly showing the presence of highest amount of toxin at Day 7 after inoculation (**D**).

**Figure 2 jof-06-00370-f002:**
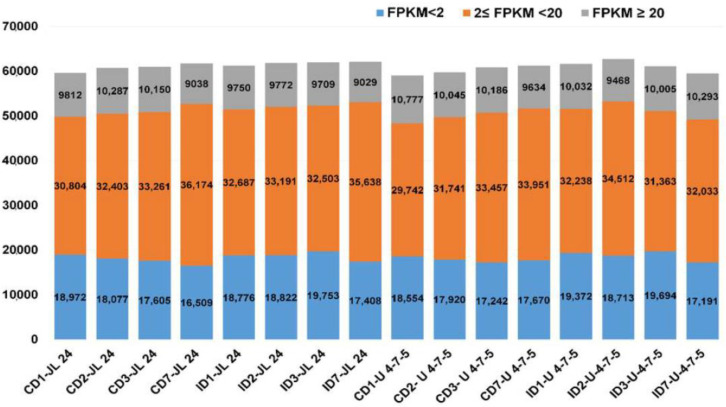
Distribution of genes expressed in JL 24 (susceptible genotype) and U 4-7-5 (resistant genotype). Three categories were grouped on the basis of Fragments Per Kilobase of exon per Million Reads Mapped (FPKM) value (FPKM < 2, 2 ≤ FPKM < 20 and FPKM ≥ 20) in 16 samples at different time points.

**Figure 3 jof-06-00370-f003:**
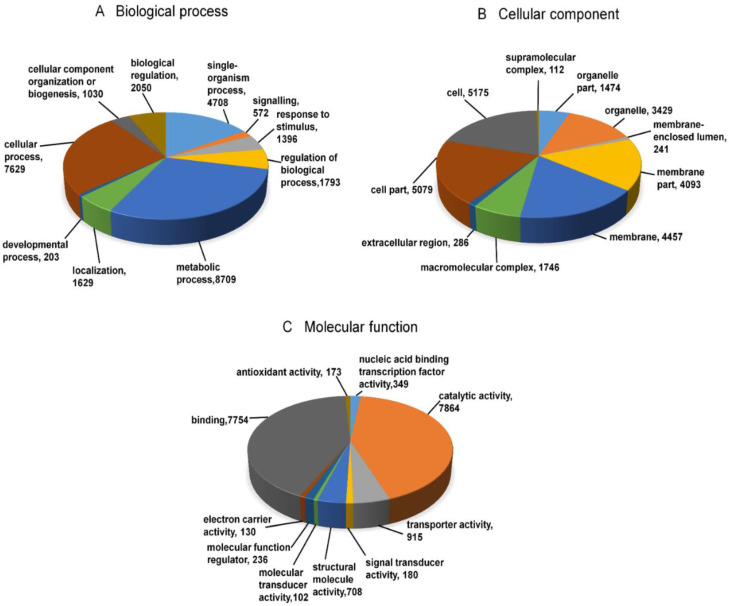
Distribution of Gene Ontology (GO) annotation assigned by blast 2 GO. This figure describes three GO categories namely, biological processes (**A**), cellular functions (**B**), and molecular functions (**C**).

**Figure 4 jof-06-00370-f004:**
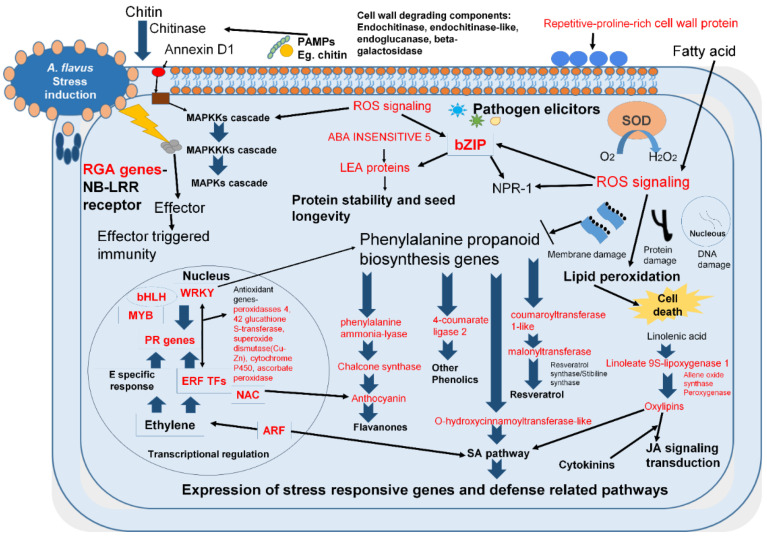
Representation of genes/pathways/transcription factors identified from host–pathogen interaction during aflatoxin production. This pathway has added an understanding of gene regulation and regulatory networks at the gene expression level. The genes/proteins/pathways/resistant factors illustrated represent the crop resistant factors that control AP contamination. The components represented in red color are the important genes and transcription factors enriched in our study. In brief, the first host–pathogen interaction occurs at the seed coat, wax and cutin which provide the first layer of defense against *A. flavus*. At the initial stage, nucleotide-binding site and leucine-rich repeats (NBS-LRR), elicitors and oxylipins play an important role in host–pathogen interactions. Transcription factors such as WRKY, bHLH, MYB, ERF, ARF trigger JA, SA and phenyl propanoid biosynthesis pathways to evoke defense responsive mechanism against *A. flavus*. In this defense responsive mechanism, PR proteins, linoleate 9S-lipoxygenase 1 protein (LOX) genes, secondary metabolites like anthocyanin, chalcone, phytoalexin compounds like resveratrol synthase/stilbene synthase are expressed during infection and play an important role at the second level of resistance mechanism. Abbreviations: JA—Jasmonic acid; SA—Salicyclic acid; ROS—Reactive oxygen species; PR—pathogenesis-related proteins; MAPK—Mitogen activated protein kinases; AFB_1_—aflatoxin B_1_; SOD—superoxide dismutase; Transcription factors—NAC (NAM, ATAF1/2 and CUC2 domain proteins), ERF—Ethylene responsive factors; NPR-1.

**Figure 5 jof-06-00370-f005:**
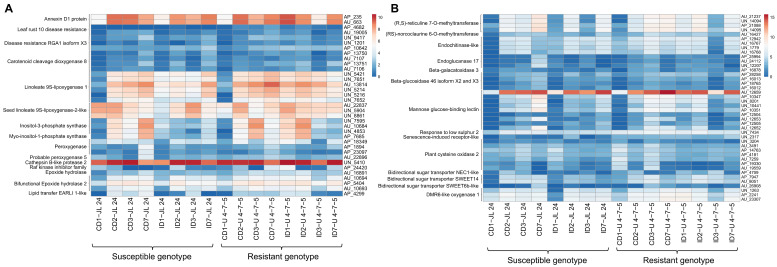
Heat maps of transcripts from disease-related pathways affected in 16 groundnut genotypes during control and infected conditions in response to *A. flavus* infection. In both (**A**,**B**), red color depicts the upregulation and blue color depicts the downregulation in the gene expression pattern. Color codes corresponds to the FPKM value of the transcripts which are increasing from blue to red color. Genotypes (JL 24 and U 4-7-5) and treatment level (control and infected) are shown at the top of the figure.

**Figure 6 jof-06-00370-f006:**
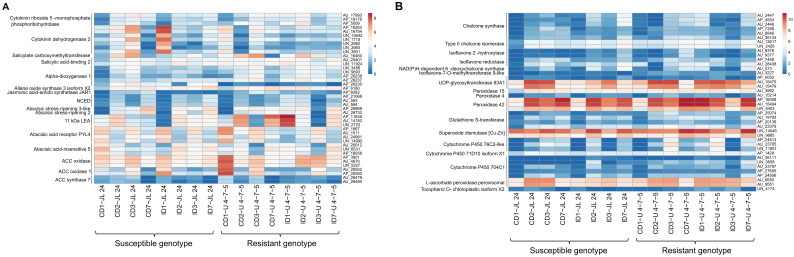
Differentially expressed genes related to hormone biosynthetic, flavonoid biosynthesis and ROS detoxifying pathways. Heat maps of (**A**) hormone biosynthetic signaling related pathways transcripts affected in 16 groundnut genotypes during control and infected conditions in response to infection with *A. flavus*. The red color depicts the upregulation and blue color depicts the downregulation in the gene expression pattern. The color code corresponds to the FPKM value of the transcripts which are increasing from blue to red color. Genotypes (JL 24 and U 4-7-5) and treatment level (control and infected) are shown at the top of the figure. Heat maps of (**B**) flavonoid biosynthesis and ROS detoxifying genes transcripts affected in 16 groundnut genotypes during control and infected conditions in response to infection with *A. flavus*.

**Figure 7 jof-06-00370-f007:**
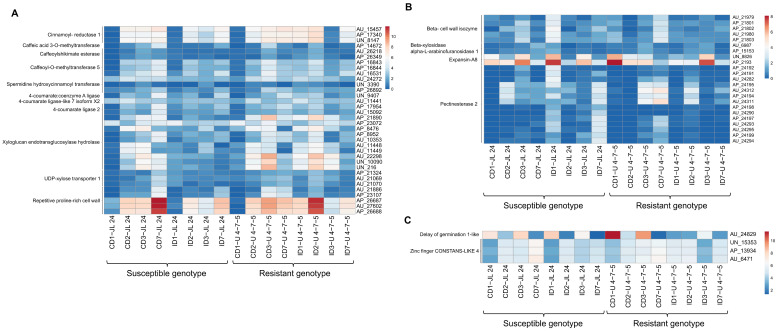
Differentially expressed genes related to cell wall and seed germination. Heat maps of (**A**) genes related to cell wall metabolism pathways in 16 groundnut genotypes during control and infected conditions in response to *A. flavus*. The red color depicts the upregulation and blue color depicts the downregulation in the gene expression pattern. The color code corresponds to the FPKM value of the transcripts which increase from yellow to purple color. Genotype and treatment level are shown at the top of the figure. Heat maps of (**B**) cell wall catabolizing gene transcripts affected in 16 groundnut genotypes during control and infected conditions in response to infection with *A. flavus*. Heat maps of (**C**) seed germination related genes transcripts affected in 16 groundnut genotypes during control and infected conditions in response to *A. flavus.*

**Table 1 jof-06-00370-t001:** Summary of sequencing data generation and mapping to transcriptome assembly.

Sample ID	Library Layout	Total Raw Reads (bp)	Total Filtered Reads (bp)	Overall Mapped Reads (bp)
CD1-U 4-7-5	76 bp	49,598,998	45,776,458	81.94%
CD2-U 4-7-5	76 bp	53,002,010	49,654,548	85.21%
CD3-U 4-7-5	76 bp	47,829,946	44,340,298	86.55%
CD7-U 4-7-5	76 bp	60,245,104	58,488,024	79.40%
ID1-U 4-7-5	76 bp	58,878,110	56,924,798	87.65%
ID2-U 4-7-5	76 bp	56,666,530	48,979,944	84.75%
ID3-U 4-7-5	76 bp	56,744,854	48,575,904	87.65%
ID7-U 4-7-5	76 bp	61,429,026	36,414,954	82.76%
CD1-JL 24	76 bp	52,011,224	43,258,504	84.50%
CD2-JL 24	76 bp	42,329,772	38,722,062	83.91%
CD3_JL 24	76 bp	42,191,954	31,798,244	81.57%
CD7-JL 24	76 bp	62,379,066	60,515,702	82.65%
ID1-JL 24	76 bp	45,726,656	36,085,764	85.34%
ID2-JL 24	76 bp	48,986,478	38,903,998	80.76%
ID3_JL 24	76 bp	59,056,330	50,464,726	83.75%
ID7-JL 24	76 bp	43,377,798	30,510,096	80.38%

**Table 2 jof-06-00370-t002:** Statistics of genome guided transcriptome assembly.

Features	Numbers/Size
Number of transcripts	74,026
Total transcript length	8,15,86,292 bp
Average transcript size	1102 bp
Transcript N50	1626 bp
Max transcript size	12,681 bp
Min transcript size	201 bp

**Table 3 jof-06-00370-t003:** Highly differentially expressed putative candidate genes for AP in groundnut identified through transcriptome profiling.

Gene ID	Days	JL 24 (I)	U 4-7-5 (I)	Log Fold	Annotation
AU_14183	Day 1	0.27	289.37	10.04	11 kDa late embryogenesis abundant
AP_26688	Day 1	0.34	170.60	8.99	repetitive proline-rich cell wall 2-like
UN_2982	Day 1	106.94	1.18	−6.51	cytokinin dehydrogenase 2
AU_24282	Day 1	12.47	0.05	−8.03	pectinesterase 2
AU_10684	Day 2	4.84	369.17	6.25	inositol-3-phosphate synthase
UN_10090	Day 2	1.56	156.62	6.65	xyloglucan endotransglucosylase hydrolase 31
AP_19330	Day 2	3.32	0.08	−5.35	plant cysteine oxidase 2 isoform X1
AU_573	Day 2	17.61	0.60	−4.87	NAD(P)H-dependent 6-deoxychalcone synthase
UN_4174	Day 3	0.12	3.09	4.70	tocopherol O-chloroplastic isoform X2
UN_216	Day 3	1.86	68.94	5.21	xyloglucan endotransglucosylase hydrolase 31
AP_11635	Day 3	22.19	0.07	−8.34	11 kDa late embryogenesis abundant
AU_15214	Day 3	2.39	0.04	−6.02	peroxidase 4
AU_3227	Day 3	0.09	1.15	3.70	isoflavone-7-O-methyltransferase 9-like
UN_216	Day 3	0.59	4.40	2.89	xyloglucan endotransglucosylase hydrolase 31
UN_10441	Day 3	25.96	4.21	−2.62	mannose glucose-specific lectin
AU_583	Day 3	10.44	2.75	−1.93	9-cis-epoxycarotenoid dioxygenase

AU tracking ID for genes identified in *A. duranensis* (A-genome) genome; AP tracking ID for genes identified in *A. ipaensis* (B-genome) genome while UN tracking ID indicates genes identified in both the genomes, NADPH: Reduced form of nicotinamide adenine dinucleotide phosphate.

**Table 4 jof-06-00370-t004:** Differentially expressed key transcripts and their putative functions.

S No.	Transcript Name	Gene ID	Putative Function	References
1.	Annexin D1 protein	AP_235 and AU_663	Defense responses	Sirko et al., 2015 [42]
2.	Disease resistance RGA1	UN_9417, UN_1201, and AP_10642	Defense responses	Sekhwal et al., 2015 [43]
3.	Linoleate 9S-lipoxygenase 1	UN_5421; UN_7651, AU_13814, UN_5214, UN_5216 and UN_7652	Stress signalling	Grebner et al., 2013 [40]
4.	JA-amino acid synthetase	AP_6282	Involved in function of jasmonic acid	Guranowski et al., 2007 [50]
5.	ABA-INSENSITIVE 5	AU_20012, UN_8531 and AP_18058	Transcription factor; Alter ABA biosynthetic genes	Finkelstein and Lynch 2000 [51]
6.	Cytochrome P450 oxidase	AU_26111, UN_3658, AU_23797, AP_27605,AP_24506	Antioxidant genes	Nayak et al., 2017 [12]
7.	Ascorbate peroxidase	AU_9550, AU_9551	Antioxidant genes	Nayak et al., 2017 [12]
8.	Seed linoleate 9S-lipoxygenase-2-like	AU_22837, UN_5904,UN_8861	Oxidation of fatty acids	Korani et al., 2018 [13]
9.	4-coumarate—ligase 2	AP_17954, AU_15092	Flavonoid biosynthesis	Wang et al., 2016 [7]

Note: AU tracking ID for genes identified in *A. duranensis* (A-genome) genome; AP tracking ID for genes identified in *A. ipaensis* (B-genome) genome while UN tracking ID indicates genes identified in both the genomes.

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
