# Peer review of "Transcriptome Analysis Identified Coordinated Control of Key Pathways Regulating Cellular Physiology and Metabolism upon Aspergillus flavus Infection Resulting in Reduced Aflatoxin Production in Groundnut"

_jof, 2020, doi:10.3390/jof6040370_

Round 1

Reviewer 1 Report

In this study, Soni et al have been carried out the transcriptome analysis of host-pathogen interactions in seeds of resistant and susceptible genotypes of groundnut resulting from Aspergillus flavus infection to decipher the molecular mechanisms of resistance to aflatoxin production. The authors discovered that differentially expressed genes involved in disease resistance, hormone biosynthetic signaling, ROS detoxifying, cell wall metabolism pathways represent the strategies of resistance to aflatoxin production. Overall, the comparative transcriptome analysis decoded the strategies of resistance to aflatoxin production at the molecular level. The understanding of host-pathogen interaction mechanisms will lead the path to develop resistance crops against Aspergillus flavus and other related fungi.

The manuscript is well written. The data are presented in a well-organized manner and discussed thoroughly. The major limitation of this study is the insufficient number of biological replicates for RNA-Seq data. The authors have used only one biological replicate for RNA-Seq data to display the host-pathogen interaction mechanisms at the molecular level. 

My main concerns are the following: 

  1. The authors have used only one biological replicate for RNA-Seq data analysis for each time point. Ideally, three Biological replicates are needed for RNA-Seq analysis. Some reports have used two biological replicates. One Biological replicate is not sufficient to conclude the statistically significant observations from RNA-Seq data. How the authors performed the statistical analysis to prove the accuracy and statistical significance of the RNA-Seq data.
  1. The authors did not validate the RNA-Seq data by using qRT-PCR of a few representative DEGs.
  2. The authors did not provide the age of seeds used in this study. That can affect the interaction with Aspergillus flavus and aflatoxin production. Are both cultivar’s seeds harvested at the same time point?

Have authors used similar size and weight of seeds for both cultivars in the study?

  1. The authors have selected the time points 24hr, 48hr, 72hr, and 7th day after inoculation for study?

What was the rationale to choose those time points?

  1. Under “Aflatoxin quantification” section, pleased provide the detail of antibodies used in this quantification. e.g. Catalog number, company name, host in which it produced, etc. Although the authors provide the reference for methodology, still a brief methodology is required here.
  2. Please provide the sample size detail for experiments included in Figure 1.

Figure 1D, how many biological replicates were used in one independent experiment. How many seeds were used in one biological replicate? How many independent experiments were carried out to confirm this observation? Please provide statistical analysis for this graph.

  1. In the Result section,

Line 171-172, “At ID3 and ID7, very low mycelial growth was observed in U 4-7-5.”

Line 172-173, “…..but the infected JL 24 failed to germinate due to fungal invasion and colonization after infection.”

However, Figure 1B showed that on the 3rd and 7th day U 4-7-5 showed more fungal infection and no germination compared to JL 24.

Please double-check this. I think JL 24 switched with U 4-7-5.

  1. Please provide the source information where you have deposited the RNA-Seq data obtained from this study.
  2. Comments for Figures 5-7

Row stands for what?

Please provide the FPKM value for minimum and maximum range scale to the color codes.

  1. The authors have used different types of abbreviations for the same term. e.g. ID or DAI or DI; 1DI or ID1 etc.

Please use the same format of abbreviation throughout the manuscript.

  1. When compare the JL24 and U 4-7-5, please also mentioned the time point.

e.g.

Line 292-294, “Further, A. flavus infection drastically increased the expression of four transcripts (AP_13750, AU_7107, AP_13751 and AU_7106) encoding for carotenoid cleavage dioxygenase 8 homolog chloroplastic protein (CCD8B gene) in U 4-7-5 compared to JL 24 (Figure 5a)”

At what time point?

Reviewer 2 Report

This article is described important information of genes expressed in JL 24 (susceptible genotype) and U 4-7-5 (resistant genotype).

Thank you for very interesting article.

I am addition comment as follows.

My comments are as follows.

Minor problem.

  1. FPKM (fragments per kilobase of exon per million reads mapped)
  2. Table 3 is wrong column in AU_583.

     3. Figure 1-A is unclearly control or infection. It has better that you write more explanation.

Author Response

This article is described important information of genes expressed in JL 24 (susceptible genotype) and U 4-7-5 (resistant genotype).

Thank you for very interesting article.

I am addition comment as follows.

Response: Authors are thankful to the Reviewer for appreciating the MS. We have responded to all comments in the following sections and revised the MS accordingly.

My comments are as follows.

Minor problem.

Point 1: FPKM (fragments per kilobase of exon per million reads mapped)

Response 1: Thanks for the comment. We have updated the same in the MS (Line 166 and 272).

Point 2: Table 3 is wrong column in AU_583.

Response 2: Thanks for bringing this formatting error. We have corrected it.

Point 3: Figure 1-A is unclearly control or infection. It has better that you write more explanation.

Response 3: Thanks for the important suggestion. We have included the revised figure with proper information.

Authors hope that the Reviewer is happy with our responses and revision. We hope that the revised version is fine with the Reviewer to recommend its acceptance. Many thanks.

Round 2

Reviewer 1 Report

Comment.v1: The authors have used only one biological replicate for RNA-Seq data analysis for each time point. Ideally, three Biological replicates are needed for RNA-Seq analysis. Some reports have used two biological replicates. One Biological replicate is not sufficient to conclude the statistically significant observations from RNA-Seq data. How the authors performed the statistical analysis to prove the accuracy and statistical significance of the RNASeq
data.

Authors response: Thanks for raising this point. We apologize to miss this point. In fact, like our earlier study Nayak et al. 2017 (https://www.nature.com/articles/s41598-017-09260-8), we also used two replicates in this study. However, as the experiment was setup and samples were collected and provided for sequencing by Spurthi Nayak, the draft of this MS was prepared by Pooja Soni (as Spurthi Nayak left ICRISAT by that time), this information couldn’t be included in the MS. We have updated the information on two biological replicates in the revised MS now as following.

“Briefly, 5μg of the total RNA pooled in equal quantity from two biological replicates were used for the construction of cDNA library using mRNA-Seq sample prep kit (Illumina Inc., USA) following manufacturer’s guidelines. Subsequentlythe RNA samples with 260/280 ratio of 1.8 to 2.1, 260/230 ratio of 2.0 to 2.3 and RIN (RNA integrity num, ber) value of >8.0, were pooled for paired-end sequencing on NextSeq 500 platform. The raw reads were subjected to quality filtering using NGSQCbox (Katta et al. 2015) [17] and Trimmomatic v0.33 (Bolger et al. 2014) [18] to remove low-quality sequencing reads with ambiguous nucleotides and any adapter contamination.”

Comment.v2: The authors pooled the total RNA from two biological replicates then used it for cDNA library construction and RNA sequencing. In this case, still, it will be considered that only one RNA sample replicate was used for RNA-Seq analysis for each time point. Since the authors have used only one replicate for RNA-Seq analysis. It became absolute mandatory to provide detailed qRT-PCR of important DEGs with three biological replicates.

I would like to recommend the qRT-PCR of highly upregulated transcription factors (ARF, DBB, MYB, NAC, C2H2) and 10% of DEGs which are discussed in results to decipher the molecular mechanisms.

We need biological replicates to carried out statistical analysis to show the statistical significance of a given observation and this is also true for RNA-Seq analysis. The minimum number of biological replicates required to generate a reliable RNA-Seq analysis depends on many factors and the heterogeneity of the biological replicates is one of them. According to the general guidelines for RNA-Seq you need at least 3 biological replicates, 4-6 is even better. Statistical significance is all about drawing interpretations about probability distributions and variance among replicates, and there are no distribution and variance of the data set with an n=1. Without distribution and variance of a given data set you can not compute a statistical significance hence can not draw any biological significance. Therefore, in the case of RNA-Seq analysis, any difference in normalized read counts for a given gene between two samples with a single replicate tells you nothing about how biologically relevant or real such difference is. 

Please read the following article to decide the minimum biological replicates required for comparative transcriptome analysis of two conditions.

Schurch et al 2016 recommended that "For future RNA-seq experiments, these results suggest that at least six biological replicates should be used, rising to at least 12 when it is important to identify SDE genes for all fold changes. If fewer than 12 replicates are used, a superior combination of true positive and false positive performances makes edgeR and DESeq2 the leading tools. For higher replicate numbers, minimizing false positives is more important and DESeq marginally outperforms the other tools."

GierliÅ„ski et al 2015 used 48 biological replicates for two conditions and concluded that it is clear that increasing the number of biological replicates above the 2 or 3 typically performed in RNA-seq experiments will be beneficial in mitigating the risk of ‘bad’ biological replicates skewing interpretation of the data."

Lamarre et al 2018 recommend at least four biological replicates per condition to be almost sure of obtaining about 1000 DEGs if they exist in real.

Reference:

Schurch, N. J., Schofield, P., Gierliński, M., Cole, C., Sherstnev, A., Singh, V., ... & Blaxter, M. (2016). How many biological replicates are needed in an RNA-seq experiment and which differential expression tool should you use?. RNA, 22(6), 839-851.

Gierliński, M., Cole, C., Schofield, P., Schurch, N. J., Sherstnev, A., Singh, V., ... & Blaxter, M. (2015). Statistical models for RNA-seq data derived from a two-condition 48-replicate experiment. Bioinformatics, 31(22), 3625-3630.

Lamarre, S., Frasse, P., Zouine, M., Labourdette, D., Sainderichin, E., Hu, G., ... & Maza, E. (2018). Optimization of an RNA-Seq differential gene expression analysis depending on biological replicate number and library size. Frontiers in plant science, 9, 108.

Author Response

Thanks for this comment. We are grateful to the Reviewer 1 for highlighting these points. At the same time we are thankful to Reviewer 2 for endorsing the MS for publication.

Regarding comment on biological replicates and qRT-PCR analysis, authors appreciate this point. At the same time, we strongly believe that these things are not essential for publication of this paper. Please note following points for your consideration:

  1. When we identify the candidate gene or identify the causal gene for the trait and develop a diagnostic maker for testing (especially in medical sciences, like marker for cancer) or undertake some gene editing work, it is really important for undertaking several biological replicates and validation work. The objective of our study is to understand the involvement of genes in molecular mechanisms of Aflatoxin production content in a crop. Therefore it is not essential for undertaking qRT PCR analysis. There are many studies, like our one, have been published earlier also that don’t carry validation work. Some of them include Espindola et al. 2018, PLoS One13(10):e0198575; Rajan & Sawant 2015, 3 Biotech5(4):585-596, etc.
  2. The Reviewer has cited some papers that are based on data modelling, simulation, etc. and not really on biological analysis. It seems that the Reviewer too has the expertise in the same area, which is good, but he/she should appreciate the applicability of these methods and limitations in crop plants. The objectives of suggested papers (and it is quite possible that the Reviewer is an author on one of these papers) is NOT identification of genes or understanding the mechanisms and they are not based on real-life experiments in crops with a genome size (in several Gbs), rather benchmarking of different tools and with different number of biological replicates. These studies indicate about 12 replicates, and probably these are possible with small genomes, yeast, etc. but I don’t think that this is possible in the crops with large genomes such as groundnut used in this study. Even I have not come across any such study that have taken 12 or 20 biological replicates for NGS and have undertaken qRT PCR for about 1/3rd of genes. This was required when we used to do Northern analysis.                                                                                 In summary, the purpose of our study and suggested papers are entirely different. And if the Reviewer would like to go ahead with published papers, then we too have cited a few papers in above point and we can give a list of some more papers where pooling of RNA samples was undertaken to understand DEGs.
  3. Having said that though it is not essential to undertake qRT PCR analysis, we could have done it provided we would have RNA samples available from those experiments. It is not wise to set us new set of experiments that may bring some variations. Moreover we also don’t have resources now- as people working on the project have already left and the project is also completed.

In view of above and as the Reviewer # 2 has not come back with any further comment, we very much hope that the Editor and Reviewer will appreciate our position on this study and also limitation not being able to carry out the qRT PCR study. We very much hope that Editor and Reviewer will consider this request and will accept our MS for publication.